# Taxonomic Characterization, and Secondary Metabolite Analysis of *Streptomyces triticiradicis* sp. nov.: A Novel Actinomycete with Antifungal Activity

**DOI:** 10.3390/microorganisms8010077

**Published:** 2020-01-05

**Authors:** Zhiyin Yu, Chuanyu Han, Bing Yu, Junwei Zhao, Yijun Yan, Shengxiong Huang, Chongxi Liu, Wensheng Xiang

**Affiliations:** 1Key Laboratory of Agricultural Microbiology of Heilongjiang Province, Northeast Agricultural University, Harbin 150030, China; zhiyinyu@yeah.net (Z.Y.); chuanyuhan@yeah.net (C.H.); yubing95yb@163.com (B.Y.); guyan2080@126.com (J.Z.); 2State Key Laboratory of Phytochemistry and Plant Resources in West China, Kunming Institute of Botany, Chinese Academy of Sciences, Kunming 650201, China; yanyijun@mail.kib.ac.cn (Y.Y.); sxhuang@mail.kib.ac.cn (S.H.); 3State Key Laboratory for Biology of Plant Diseases and Insect Pests, Institute of Plant Protection, Chinese Academy of Agricultural Sciences, Beijing 100193, China

**Keywords:** *Streptomyces triticiradicis* sp. nov., antifungal activity, rhizosphere soil, new compound, genome analysis

## Abstract

The rhizosphere, an important battleground between beneficial microbes and pathogens, is usually considered to be a good source for isolation of antagonistic microorganisms. In this study, a novel actinobacteria with broad-spectrum antifungal activity, designated strain NEAU-H2^T^, was isolated from the rhizosphere soil of wheat (*Triticum aestivum* L.). 16S rRNA gene sequence similarity studies showed that strain NEAU-H2^T^ belonged to the genus *Streptomyces*, with high sequence similarities to *Streptomyces rhizosphaerihabitans* NBRC 109807^T^ (98.8%), *Streptomyces populi* A249^T^ (98.6%), and *Streptomyces siamensis* NBRC 108799^T^ (98.6%). Phylogenetic analysis based on 16S rRNA, *atpD*, *gyrB*, *recA*, *rpoB*, and *trpB* gene sequences showed that the strain formed a stable clade with *S. populi* A249^T^. Morphological and chemotaxonomic characteristics of the strain coincided with members of the genus *Streptomyces*. A combination of DNA–DNA hybridization results and phenotypic properties indicated that the strain could be distinguished from the abovementioned strains. Thus, strain NEAU-H2^T^ belongs to a novel species in the genus *Streptomyces*, for which the name *Streptomyces triticiradicis* sp. nov. is proposed. In addition, the metabolites isolated from cultures of strain NEAU-H2^T^ were characterized by nuclear magnetic resonance (NMR) and mass spectrometry (MS) analyses. One new compound and three known congeners were isolated. Further, genome analysis revealed that the strain harbored diverse biosynthetic potential, and one cluster showing 63% similarity to natamycin biosynthetic gene cluster may contribute to the antifungal activity. The type strain is NEAU-H2^T^ (= CCTCC AA 2018031^T^ = DSM 109825^T^).

## 1. Introduction

It is well known that plant pathogenic fungi can cause a tremendous loss of global agricultural production [1]. Despite synthetic fungicides being effective and playing an indispensable role against pathogenic fungi, the available antifungal agents are far from satisfactory as a result of several drawbacks, such as severe drug resistance, drug-related toxicity, and many other problems [2]. Therefore, novel antifungal agents and antagonistic microorganisms are needed to effectively control the fungal diseases of agricultural crops. Natural products and their derivatives, in particular secondary metabolites derived from *Streptomyces*, have always been valuable sources for lead discovery in medicinal and agricultural chemistry because their novel scaffolds can provide new modes of action [3,4]. Members of the genus *Streptomyces* species are gram-positive, filamentous, and sporulating actinobacteria containing a number of biosynthetic gene clusters, indicating their potential ability to produce large numbers of secondary metabolites with diverse biological activities [5,6], and they represent the source of 75% of clinically useful antibiotics presently available [7]. Many *Streptomyces* species have been successfully developed as commercial biofungicides based on *Streptomyces griseoviridis* [8]. Thus, *Streptomyces* are still an attractive and indispensable resource for drug discovery.

The rhizosphere is an environment where pathogenic and beneficial microorganisms constitute a major influential force on plant growth and health, which differs from the bulk soil [9]. Plants not only provide nutrients for microbial growth but also change the microbial diversity and increase the numbers of bioactive microorganisms in the rhizosphere. The rhizosphere provides an excellent place for pursuing actinobacteria producing novel antibiotics [10,11]. During our search for antagonistic actinobacteria from the rhizosphere soil of wheat (*Triticum aestivum* L.), the strain NEAU-H2^T^ was isolated, which showed broad inhibitory activities against phytopathogenic fungi. Based on the polyphasic taxonomy analysis, this strain was classified as representative of a novel species in the genus *Streptomyces.* In addition, the secondary metabolites of this strain were investigated by spectroscopic and genomic analyses.

## 2. Materials and Methods

### 2.1. Isolation of Actinobacterial Strain

Strain NEAU-H2^T^ was isolated from the rhizosphere soil of wheat (*Triticum aestivum* L.) collected from Zhumadian, Henan Province, Central China (32°98′ N, 114°02′ E). The root sample was air-dried for 24 h at room temperature, and then the surface soil was shaken off gently. After, the sample was shaken at 250 rpm in 100 mL of sterile water with glass beads for 30 min at 20 °C and then filtered with a single layer of gauze to obtain the rhizosphere soil suspension. The suspension was serially diluted and spread on cellulose-proline agar (CPA) [12] supplemented with cycloheximide (50 mg·L^−1^) and nalidixic acid (20 mg·L^−1^), and cultured at 28 °C for 3 weeks. Strain NEAU-H2^T^ was isolated and purified on the International *Streptomyces* Project (ISP) medium 3 [13], and maintained as glycerol suspensions (20%, *v*/*v*) at −80 °C.

### 2.2. Morphological and Biochemical Characteristics

Gram staining was performed by the Hucker method [14]. Morphological characteristics were observed by light microscopy (Nikon ECLIPSE E200, Nikon Corporation, Tokyo, Japan) and scanning electron microscopy (Hitachi SU8010, Hitachi Co., Tokyo, Japan) using cultures grown on ISP 3 agar at 28 °C for 2 weeks. Samples for scanning electron microscopy were prepared as described by Jin et al. [15]. Cultural characteristics were determined on the ISP media 1–7 [13], Bennett’s agar [16], Czapek’s agar [17], and Nutrient agar [18] after 2 weeks at 28 °C. The color of substrate mycelium, aerial mycelium, and diffusible pigment on the different tested media were determined using color chips from the ISCC-NBS color charts [19]. Temperature tolerance for growth was evaluated at 4, 10, 15, 20, 25, 28, 35, 37, 40, and 45 °C on ISP 3 agar after incubation for 2 weeks. The pH range for growth (pH 3.0–12.0, at intervals of 1.0 pH unit) was tested in GY broth [20] using the buffer system described by Zhao et al. [21] and NaCl tolerance (0%–15% (*w*/*v*) in 1% intervals) for growth was determined after 2 weeks growth in GY broth at 28 °C with shaking at 250 rpm. Hydrolysis of Tweens (20, 40, and 80) and production of urease were tested according to the method of Smibert and Krieg [14]. The utilization of sole carbon and nitrogen sources were determined following the methods of Gordon et al. [22]. The decomposition of cellulose, hydrolysis of starch, coagulation of milk, aesculin, reduction of nitrate, liquefaction of gelatin, and production of H_2_S were examined as described previously [23].

### 2.3. Chemotaxonomic Analysis

The freeze-dried cells used for chemotaxonomic analysis were obtained from cultures grown in GY medium on a rotary shaker for seven days at 28 °C. Cells were acquired and washed twice with sterile distilled water and freeze-dried. The isomer of diaminopimelic acid (DAP) in the cell wall hydrolysates was derivatized and analyzed by HPLC (Agilent TC-C_18_ Column, 250 × 4.6 mm, i.d. 5 μm) with a mobile phase consisting of acetonitrile/phosphate buffer (0.05 mol·L^−1^, pH 7.2, 15:85, *v*/*v*), and a flow rate of 0.5 mL·min^−1^ at a column temperature of 28 °C [24]. An Agilent G1321A fluorescence detector was used to detect the peak with a 365 nm excitation and 455 nm longpass emission filters. The whole-cell sugars were analyzed according to Lechevalier [25]. The polar lipids were extracted and examined by two-dimensional TLC (thin-layer chromatography, Qingdao Marine Chemical Inc., Qingdao, China) and identified according to the method of Minnikin et al. [26]. Menaquinones were extracted and purified from freeze-dried biomass following the methods of Collins [27]. The extracts were analyzed by HPLC-UV (Agilent Extend-C_18_ Column, 150 × 4.6 mm, i.d. 5 μm, 1.0 mL·min^−1^ acetonitrile: iso-propyl alcohol = 60:40) at 270 nm [28]. Fatty acid methyl esters were performed by GC-MS according to the method of Xiang et al. [29] and identified with the NIST 14 database.

### 2.4. Phylogenetic Analysis 

Strain NEAU-H2^T^ was grown on ISP 3 agar plates for one week at 28 °C. Then, it was inoculated into 250-mL baffle Erlenmeyer flasks containing 50 mL of GY broth and cultivated for two days at 28 °C with shaking at 250 rpm. After that, the total DNA was extracted according to the lysozyme-sodium dodecyl sulfate-phenol/chloroform method [30]. The primers and procedure for PCR amplification were carried out as described by Yi et al. [31]. The PCR product was purified and cloned into the vector pMD19-T (Takara, Shiga, Japan) and sequenced using an Applied Biosystems DNA sequencer (model 3730XL, Applied Biosystems Inc., Foster City, CA, USA). Almost full-length 16S rRNA gene sequence (1519 bp) was multiply aligned in MEGA (Molecular Evolutionary Genetics Analysis) using the Clustal W algorithm and trimmed manually if necessary. Phylogenetic trees were constructed with neighbor-joining [32] and maximum likelihood [33] algorithms using MEGA software version 7.0 (Kumar S, Philly, PA, USA) [34]. The stability of the topology of the phylogenetic tree was assessed using the bootstrap method with 1000 repetitions [35]. A distance matrix was calculated using Kimura’s two-parameter model [36]. All positions containing gaps and missing data were eliminated from the dataset (complete deletion option). The calculation of 16S rRNA gene sequence similarities between strains was carried out on the basis of pairwise alignment using the EzBioCloud server (https://www.ezbiocloud.net/) [37]. Phylogenetic relationships of strain NEAU-H2^T^ were also confirmed using sequences for five individual housekeeping genes (*atpD*, *gyrB*, *recA*, *rpoB*, and *trpB*). These sequences of housekeeping genes of strain NEAU-H2^T^ were obtained from the Whole Genome sequences. The sequences of each locus were aligned using the software package MEGA version 7.0 and trimmed manually at the same position before being used for further analysis. Phylogenetic analysis was performed as described above.

### 2.5. DNA–DNA Relatedness Tests 

The total DNA was extracted according to the method in the Section 2.4. The harvested DNA was detected by agarose gel electrophoresis and quantified by a Qubit 2.0 Fluorometer (Thermo Scientific, Ashville, NC, USA). The Illumina Novaseq PE150 (Illumina, San Diego, CA, USA) platform was used to perform whole-genome sequencing. A-tailed, ligated to paired-end adaptors, and PCR-amplified samples with a 350-bp insert were used for the library construction at the Beijing Novogene Bioinformatics Technology Co., Ltd. Illumina PCR adapter reads and low-quality reads from the paired end were filtered with a quality control step by our own compling pipeline. All good-quality paired reads were assembled by the SOAP (Short Oligonucleotide Alignment Program) denovo [38,39] (https://github.com/aquaskyline) into a number of contigs. After that, the filter reads were handled by the next step of the gap closing. 

Because of a lack of the whole genome sequence of strains *Streptomyces rhizosphaerihabitans* NBRC 109807^T^ and *Streptomyces siamensis* NBRC 108799^T^, a DNA–DNA relatedness test was carried out as described by De Ley et al. [40] under consideration of modifications [41] with a model Cary 100 Bio UV/VIS-spectrophotometer (Hitachi U-3900, Hitachi Co., Tokyo, Japan) and a temperature controller. The DNA hybridization samples were diluted to OD_260_ around 1.0 using 0.1 × SSC (saline sodium citrate buffer), then sheared using a JY92-II ultrasonic cell disruptor (ultrasonic time 3s, interval time 4 s, 90 times). The DNA renaturation rates were measured in 2 × SSC at 70 °C. This experiment was repeated three times to calculate the average value. The DNA–DNA relatedness value was determined between the genomes of strain NEAU-H2^T^ and *Streptomyces populi* A249^T^ (PJOS01000000) using the genome-to-genome distance calculator (GGDC 2.0) at http://ggdc.dsmz.de [42]. Genome mining analysis was performed with antiSMASH (version 4.0, Blin K, Oxford, UK) [43].

### 2.6. In Vitro Antifungal Activity Test

Antifungal screening was performed against 10 different phytopathogenic fungi: *Sclerotinia sclerotiorum*, *Exserohilum turcicum*, *Colletotrichum orbiculare*, *Corynespora cassiicola*, *Rhizoctonia solani*, *Fusarium graminearum*, *Fusarium oxysporum*, *Sphacelotheca reiliana*, *Curvularia lunata*, and *Helminthosporium maydis*. Ten phytopathogenic fungi were preserved in the Key Laboratory of Agricultural Microbiology within the Heilongjiang province, China. Antifungal activity of strain NEAU-H2^T^ was assessed using the dual culture plate assay [44]. The strain was point-inoculated at the margin of potato dextrose agar (PDA) [45] plates and cultivated for three days at 28 °C, after which a fresh mycelial PDA agar plug of the fungus was transferred into the opposite margin of the corresponding plate. Inhibition of hyphal growth of the fungus was recorded after incubated for seven days at 28 °C. The percentage inhibition rates were calculated using the formula: Inhibition rate (%) = Wi/W × 100%, where Wi is the width of inhibition and W is the width between the pathogen and actinobacteria. The assay was repeated three times and the average was calculated.

### 2.7. Isolation and Characterization of Secondary Metabolites

Strain NEAU-H2^T^ was grown on ISP 3 agar plates for five days at 28 °C. Then, it was inoculated into 250-mL baffle Erlenmeyer flasks containing 50 mL of tryptone soy broth (TSB) and cultivated for one day at 28 °C with shaking at 250 rpm. After that, aliquots (15 mL) of the culture were transferred into 1-L baffled Erlenmeyer flasks filled with 250 mL of the production medium (tryptone 0.1%, glucose 3%, beef extract 0.5%, 0.25% CaCO_3_, 0.5% NaCl, 0.1% minor elements concentrate (FeSO_4_·7H_2_O 1.0 g, CuSO_4_·5H_2_O 0.45 g, ZnSO_4_·7H_2_O 1.0 g, MnSO_4_·4H_2_O 0.1 g, K_2_MoO_4_ 0.1 g, distilled water 1 L), pH 7.2–7.4), and cultured at 30 °C for six days with shaking at 250 rpm. 

The fermentation broth (25 L) was centrifuged (4000 rev/min, 20 min), and the supernatant was extracted with ethylacetate three times. The ethylacetate extract was evaporated under reduced pressure at temperatures within 40 °C to yield an oily crude extract (5.0 g). The mycelia were extracted with methanol (1 L) and then concentrated in vacuo to remove the methanol to yield the aqueous concentrate. The mycelia concentrate was extracted with ethylacetate (1 L × 3) to afford 1.0 g of crude extract after removing the ethylacetate. Both extracts displayed most of the similar secondary metabolites based on HPLC analyses. Thus, they were combined for further purification. The samples were applied to reverse-phase HPLC analysis eluted with a flow rate of 1 mL·min^−1^ over a 28 min gradient with water and methanol (T = 0 min, 10% methanol; T = 20.0 min, 100% methanol; T = 24.0 min, 100% methanol; T = 24.1 min, 10% methanol; T = 28.0 min, 10% methanol) and at 25 °C.

The crude extract (6.1 g) was subjected to silica gel CC using a successive elution of petroleum ether/ethylacetate (1:0, 10:1, 5:1, 1:1 and 0:1, *v*/*v*) to yield A−F fractions. Fraction D (petroleum ether/ethylacetate = 1:1, *v*/*v*) was subjected to semipreparative HPLC (YMC- Hydrosphere C_18_ column, 250 mm × 10 mm i.d., 5 μm, 0–20.0 min CH_3_OH:H_2_O = 35:55, *v*/*v*, 3 mL/min) to afford **1** (*t*_R_ = 25.4 min, 2.2 mg) and **2** (*t*_R_ = 30.4 min, 2.6 mg). Compound **4** (*t*_R_ = 17.2 min, 5.5 mg) was obtained from the fraction C (petroleum ether/ethylacetate = 5:1, *v*/*v*) by semipreparative HPLC (YMC-Triart C_18_ column, 250 mm × 10 mm i.d., 5 μm, CH_3_OH:H_2_O = 25:75, *v*/*v*, 3 mL/min). Fraction E (petroleum ether/ethylacetate = 0:1, *v*/*v*) was further purified by semipreparative HPLC (YMC-Hydrosphere C_18_ column, 250 mm × 10 mm i.d., 5 μm, 0–15.0 min, CH_3_OH:H_2_O = 25:75; 15.1–30 min, CH_3_OH:H_2_O = 35:65, *v*/*v*, 3 mL/min) to give compound **3** (*t*_R_ = 21.4 min, 2.0 mg). 

NMR spectra were recorded in methanol-d4 or DMSO-d6 using a Bruker AVANCE III-600 or a Bruker AVANCE III-400 spectrometer (Bruker Corp., Karlsruhe, Germany) and TMS was used as the internal standard. HR-ESI-MS data were obtained using an Agilent G6230 Q-TOF mass instrument (Agilent Technologies Inc. Santa Clara, CA, USA). Thin-layer chromatography (TLC) was performed using precoated silica gel GF254 plates (Qingdao Marine Chemical Inc., Qingdao, China), and spots were visualized by UV light (254 nm) and colored by iodine, or by spraying heated silica gel plates with 10% H_2_SO_4_ in ethanol. Semipreparative HPLC was conducted on a HITACHI Chromaster system (Hitachi-DAD, Tokyo, Japan).

### 2.8. In Vitro Antifungal Activity Test of Compounds

The fungi were retrieved from the storage tube and cultured for seven days at 28 °C on PDA. All fungi were further cultured for one week to get new mycelium for the antifungal assays in PDA at 28 °C. The medium was mixed with the pathogenic fungi suspension at about 45 °C, ensuring the abundance of the strains was about 10^8^ cfu/mL. Next, the mixture was poured on 9-cm Petri dishes. The tested compounds were dissolved in DMSO at a concentration of 2 mg/mL. Each filter paper (5 mm in diameter) was impregnated with 10 μL of the tested compounds. The inoculated Petri dishes were cultured for 3 to 4 d at 28 °C. DMSO served as a blank control. The assay was measured three times. The inhibition diameter was measured by the cross bracketing method [46].

## 3. Results and Discussion

### 3.1. Polyphasic Taxonomic Characterization of NEAU-H2^T^

Morphological observation of the two-week culture of strain NEAU-H2^T^ grown on ISP 3 medium revealed that it had characteristics typical of the genus *Streptomyces* [47]. Aerial and substrate mycelium were well developed without fragmentation. Spiral spore chains with spiny surfaced spores (0.8–1.0 × 1.0–1.3 μm) were borne on the aerial mycelium (Figure 1). Strain NEAU-H2^T^ exhibited good growth on ISP 1–4, ISP 7, and Nutrition agar media; moderate growth on Bennett’s and Czapek’s agar media; and poor growth on ISP 5 and ISP 6 media. The colony colors varied from white to moderate yellow. A dark grayish olive pigment was produced on ISP 6 medium. The detailed cultural characteristics of strain NEAU-H2^T^ are shown in Appendix A. Strain NEAU-H2^T^ was found to grow at a temperature range of 4 to 40 °C (optimum temperature 28 °C), pH 5 to 10 (optimum pH 7), and NaCl tolerance of 0% to 9% (optimum NaCl of 0% to 1%). The physiological and biochemical properties of strain NEAU-H2^T^ are given in Table 1 and the species description.

Chemotaxonomic analyses revealed that strain NEAU-H2^T^ also exhibited the typical characteristics of the genus *Streptomyces* [47]. It contained LL-diaminopimelic acid as cell wall diamino acid, indicating that the strain is of cell wall chemotype I [51]. The whole-cell sugar was found to contain glucose. The phospholipid profile consisted of diphosphatidylglycerol (DPG), phosphatidylethanolamine (PE), phosphatidylinositol (PI), phosphatidylinositolmannosides (PIM), and an unidentified phospholipid (PL), corresponding to phospholipid type II [52] (Appendix A). The major cellular fatty acids (>10%) were iso-C_16:0_ (21.6%), anteiso-C_15:0_ (19.4%), iso-C_15:0_ (16.9%), and anteiso-C_17:0_ (13.0 %), which is fatty acid type IIc [53]; minor amounts of C_16:1_ω7c (8.5 %), C_16:0_ (7.5 %), iso-C_14:0_ (7.2%), C_17:0_ cyclo (2.1%), C_17:1_ω8c (2.1%), C_18:0_ (1.2%), and C_15:0_ (0.5%) were also present. The menaquinones detected were MK-9(H_8_) (57.5%), MK-9(H_6_) (32.3%), and MK-9(H_4_) (10.2%), which have been reported for most species of the genus *Streptomyces* [47]. 

EzBioCloud analysis suggests that strain NEAU-H2^T^ belongs to the genus *Streptomyces*. The novel strain shared the highest 16S rRNA gene sequence similarities with *S. rhizosphaerihabitans* NBRC 109807^T^ (98.8%), *S. populi* A249^T^ (98.6%), and *S. siamensis* NBRC 108799^T^ (98.6%). In the neighbor-joining phylogenetic tree based on 16S rRNA gene sequences, strain NEAU-H2^T^ formed a separate clade with *S. populi* A249^T^ (Figure 2), a relationship also recovered by the maximum likelihood algorithm (Appendix A). To further clarify the affiliation of strain NEAU-H2^T^ to closely related strains, phylogenetic trees were constructed from the concatenated sequence alignment of the five housekeeping genes based on the neighbor-joining and maximum likelihood algorithms (Figure 3 and Appendix A), which showed the same topology as the 16S rRNA gene tree. Furthermore, the concatenated sequences of *atp*D-*gyr*B-*rec*A-*rpo*B-*trp*B were used to calculate pairwise distances well above 0.007 (Appendix A) for the related species, which was considered to be the threshold for species determination [54]. Based on the 16S rRNA gene sequence similarities and phylogenetic trees, *S. rhizosphaerihabitans* NBRC 109807^T^, *S. populi* A249^T^, and *S. siamensis* NBRC 108799^T^ were selected as the closely related strains for subsequent comparative analysis.

The assembled genome sequence of strain NEAU-H2^T^ was found to be 9,921,301 bp long and composed of 135 contigs with an N50 of 167,996 bp, a DNA G+C content of 71.5 mol%, and a coverage of 152.0×. It was deposited into GenBank under the accession number WBKG00000000. Detailed genomic information is presented in the Appendix A. DNA–DNA hybridization was employed to further clarify the relatedness between strain NEAU-H2^T^ and *S. rhizosphaerihabitans* NBRC 109807^T^ and *S. siamensis* NBRC 108799^T^. The DNA–DNA relatedness values were 33.3 ± 2.5% and 44.5 ± 3.5%, respectively. Digital DNA–DNA hybridization was employed to clarify the relatedness between strain NEAU-H2^T^ and *S. populi* A249^T^. The level of DNA–DNA relatedness between them was 56.5 to 62.1%. According to the description proposed by Wayne et al. [55], the relatedness values are below the threshold value of 70% for assigning bacterial strains to the same genomic species.

Besides the genotypic evidence above, some obvious differences can also be found between strain NEAU-H2^T^ with its closely related strains regarding several phenotypic and chemotaxonomic characteristics. Strain NEAU-H2^T^ could be easily distinguished from its most closely related species by cultural characteristics, such as colony colors and diffusible pigment production (Appendix A). Morphological characteristics, including spore chain and surface ornamentation, could also distinguish the isolate from its closely related strains (Table 1). In addition, the isolate was able to grow at 4 °C, in contrast to its closely related strains, which could not. The novel strain could not utilize l-serine, l-tyrosine, l-arabinose, and meso-inositol while the closely related species could. Strain NEAU-H2^T^ was found to contain both PI and PIM in its phospholipid profile, which could distinguish it from *S. siamensis* NBRC 108799^T^ and *S. populi* A249^T^. The presence of MK-9(H_4_) could differentiate the isolate from *S. rhizosphaerihabitans* NBRC 109807^T^. Most notably, the whole-cell sugar of strain NEAU-H2^T^ was evidently different from that of *S. rhizosphaerihabitans* NBRC 109807^T^ and *S. populi* A249^T^, with the only presence of glucose. The detailed characteristics of strain NEAU-H2^T^ in comparison with its closely related strains are listed in Table 1.

Therefore, it is evident from the genotypic, phenotypic, and chemotaxonomic data that strain NEAU-H2^T^ represents a novel species of the genus *Streptomyces*, for which the name *Streptomyces triticiradicis* sp. nov. is proposed.

### 3.2. Description of Streptomyces triticiradicis sp. nov.

*Streptomyces triticiradicis* (tri.ti.ci.ra’di.cis. L. neut. n. *triticum* wheat; L. fem. n. *radix* a root; N.L. gen. n. *triticiradicis* of a wheat root).

This is an aerobic gram-staining-positive actinomycete that forms well-developed, branched substrate hyphae and aerial mycelium that differentiate into spiral spore chains consisting of spiny surfaced spores. It has good growth on ISP 1–4, ISP 7 and Nutrient agar media, moderate growth on Bennett’s and Czapek’s agar media, and poor growth on ISP 5 and ISP 6 media. A dark grayish olive pigment is produced on ISP 6 medium. Growth is observed at temperatures between 4 and 40 °C, with an optimum temperature of 28 °C. Growth occurs in the pH range from 5.0 to 10.0 (optimum pH 7.0) with 0% to 9.0% (*w*/*v*) NaCl tolerance (optimum 0%–1%). It is positive for coagulation of milk; decomposition of cellulose; hydrolysis of aesculin, starch, and Tweens (20, 40, and 80); and production of H_2_S and urease; but negative for liquefaction of gelatin, production of catalase, and reduction of nitrate. l-alanine, l-arginine, l-asparagine, l-aspartic acid, creatine, l-glutamic acid, l-glutamine glycine, l-proline, and l-threonine are utilized as sole nitrogen sources but not l-serine or l-tyrosine. d-Fructose, d-galactose, d-glucose, lactose, d-maltose, d-mannose, d-raffinose, l-rhamnose, and d-sucrose are utilized as sole carbon sources but not l-arabinose, dulcitol, meso-inositol, d-mannitol, d-ribose, d-sorbitol, or d-xylose. The cell wall contains LL-diaminopimelic acid as diagnostic diamino acid and the whole cell hydrolysate contains glucose. The major menaquinones are MK-9(H_8_), MK-9(H_6_), and MK-9(H_4_). The polar lipids profile contains DPG, PE, PI, PIM, and PL. Major fatty acids (>10%) are iso-C_16:0_, anteiso-C_15:0_, iso-C_15:0_, and anteiso-C_17:0_. The DNA G + C content of the type strain is 71.5 mol%.

The type strain is NEAU-H2^T^ (= CCTCC AA 2018031^T^ = DSM 109825^T^), isolated from the rhizosphere soil of wheat (*Triticum aestivum* L.) collected from Zhumadian, Henan Province, Central China. The GenBank/EMBL/DDBJ accession number for the 16S rRNA gene sequence of strain NEAU-H2^T^ is MN512450. This Whole Genome Shotgun project has been deposited at DDBJ/ENA/GenBank under the accession WBKG00000000. The version described in this paper is version WBKG01000000.1.

### 3.3. Antifungal Activity Evaluation

Strain NEAU-H2^T^ showed a wide range of inhibitory effects on the mycelial growth of the 10 tested phytopathogenic fungi (Figure 4A). It displayed significant inhibitory effects against four phytopathogenic fungi, including *C. orbiculare*, *C. cassiicola*, *S. sclerotiorum*, and *E. turcicum*, with the inhibition rate ranging from 48.4% to 69.0% (Figure 4B).

### 3.4. Identified of Secondary Metabolites from Strain NEAU-H2^T^

Only major components were identified from the liquid fermentation extract. Compound **1** was obtained as white amorphous powder, and its molecular formula, C_12_H_13_NO_3_, was determined by high resolution electrospray ionization mass spectrometry (HRESIMS) data (m/z 242.0792 [M + Na]^+^, calculatedd for 242.0788), corresponding to 7 degrees of unsaturation (Appendix A). The ^1^H NMR showed the presence of five aromatic protons with signals at *δ*_H_ 8.29 (s, 1H), 8.27 (d, *J* = 7.3 Hz, 1H), 7.46 (d, *J* = 7.4 Hz, 1H), 7.24 (td, *J* = 7.2, 1.2 Hz, 1H), and 7.21 (td, *J* = 7.2, 1.2 Hz, 1H), which indicated a three-substituted indole moiety (Table 2, Appendix A). The ^13^C NMR and HSQC spectra revealed 12 carbons, which were classified into one methyl (*δ*_C_ 17.9), five sp^2^ methines (*δ*_C_ 135.8, 124.5, 123.4, 122.9, 112.9), three sp^2^ quaternary carbons (*δ*_C_ 138.2, 127.3, 116.3), and two oxygenated tertiary carbons (*δ*_C_ 79.7 and 71.1) and a carbonyl carbon (*δ*_C_ 197.2) (Appendix A). 

The ^1^H-^1^H COSY and HSQC spectra of **1** showed two spin-coupling systems, H-9/H-10/H-11 and H-4/H-5/H-6/H-7 (Figure 5B). The HMBC cross-peaks from H-5 to C-3a, from H-6 to C-7a, and from H-2 to C-3/3a/C-7a further revealed the presence of an indole moiety. Cross-peaks from H-9 to C-8 and from H-10 to C-8 were observed in the HMBC spectrum, which suggested a 2,3-dihydroxybutanone connected with indole moiety at C-3 (Figure 5B). Therefore, the planar structure **1** was elucidated as depicted in Figure 5A.

Compound **2** was isolated as a colorless powder, HRESIMS m/z 235.0532 [M + Na]^+^(calculated for C_9_H_12_N_2_O_2_S, 235.0512); ^1^H NMR data (600 MHz) *δ*_H_ 7.03 (1H, dd, *J* = 2.4, 1.4 Hz, H-7), 6.96 (1H, dd, *J* = 3.9, 1.3 Hz, H-9), 6.20 (1H, dd, *J* = 3.8, 2.5 Hz, H-8), 3.13 (2H, t, *J* = 6.7 Hz, H-4), 3.38 (2H, t, *J* = 6.7 Hz, H-3), 1.92 (3H, s, H-1); ^13^C NMR data (150 MHz, CD_3_OD) *δ*_C_ 181.9 (C-5), 173.5 (C-2), 131.2 (C-6), 125.7 (C-7), 116.4 (C-9), 111.1 (C-8), 40.7 (C-3), 28.2 (C-4), and 22.5 (C-1) (Appendix A). Compound **2** was proven to be 3-Acetylamino-N-2-thienyl-propanamide by direct comparison of these data with those from the literature [56].

Compound **3**: ^1^H NMR data (600 MHz, DMSO-*d*_6_) *δ*_H_ 8.37 (1H, s, H-8), 8.21 (1H, s, H-2), 5.90 (1H, d, *J* = 5.9 Hz, H-1′), 4.55 (2H, t, *J* = 5.4 Hz, H-2′), 4.14 (1H, m, H-3′), 3.95 (1H, q, *J* = 3.3 Hz, H-4′), 3.67 (1H, dd, *J* = 12.1, 3.5 Hz, H-5′), 3.55 (1H, dd, *J* = 12.1, 3.5 Hz, H-5′); ^13^C NMR data (150 MHz, DMSO-*d*_6_) *δ*_C_ 154.3 (C-6), 151.7 (C-2), 149.9 (C-4), 138.6 (C-8), 119.8 (C-5), 87.8 (C-1′), 85.8 (C-4′), 73.6 (C-2′), 70.5 (C-3′), and 61.6 (C-5′) (Appendix A). Compound **3** was proven to be *ß*-adenosine by direct comparison of these data with those from the literature [57].

Compound **4**: ^1^H NMR data (400 MHz, CD_3_OD) *δ*_H_ 6.94 (1H, s, H-4), 6.85 (1H, d, *J* = 2.7 Hz, H-2), 6.18 (1H, m, H-3); ^13^C NMR (100 MHz, CD_3_OD) *δ*_C_164.6 (C-6), 124.4 (C-2), 124.1 (C-5), 116.6 (C-4), and 110.6 (C-3) (Appendix A). Compound **4** was proven to be 2-minaline by direct comparison of these data with those from the literature [58].

### 3.5. Mining the Biosynthetic Potential of the Strain

All the compounds were evaluated for their antifungal activity, which showed no significant inhibitory activity. In order to further discover the biosynthetic potential of the strain, we performed draft genome sequencing analysis. AntiSMASH analysis led to the identification of 38 putative gene clusters in the genome of strain NEAU-H2^T^. Eleven clusters were identified belonging to a family of polyketide synthases (PKSs), including four type I PKSs, one type II PKS, and three type III PKSs. Likewise, further genome sequence analysis revealed eight additional gene clusters comprising modular enzyme coding genes, such as non-ribosomal peptide synthetase (NRPS, four clusters) and hybrid PKS-NRPS genes (four clusters). Other gene clusters included seven terpene gene clusters, three bacteriocin gene clusters, two siderophore gene clusters, one lanthipeptide gene cluster, one lassopeptide gene cluster, one melanin gene cluster, one ectoine gene cluster, and three butyrolactone gene clusters. 

The important feature of NRPs is their ability to use nonproteinogenic amino acids as building blocks. By using such building blocks, NRPSs are able to produce peptides with diverse structures and bioactivities. As such, many NRPSs have been developed into pharmaceuticals, such as vancomycin, daptomycin, and β-lactam [59].

However, only a few metabolites were isolated and identified in culture broth under laboratory conditions from strain NEAU-H2^T^. One answer is that most biosynthetic gene clusters of secondary metabolites are cryptic in culture broth under conventional laboratory culture conditions [60]. In addition, an active ingredient has not been isolated, possibly due to the low production of these metabolites under our culture conditions.

One of the secondary metabolite biosynthetic gene clusters of strain NEAU-H2^T^ shows a 63% similarity to the biosynthetic gene cluster of natamycin, which is a 26-membered polyene macrolide antifungal agent produced by *Streptomyces chattanoogensis* L10, and the macrolide core was synthesized by five PKSs (ScnS0, ScnS1, ScnS2, ScnS3, and ScnS4) in turn [61]. Natamycin is currently widely used as an antifungal agent in human therapy and the food industry [62]. However, considering the poor quality of the genome sequence, with a large number of contigs, this may not be related to the antifungal active components identified with antibiotics and secondary metabolite analysis shell–antiSMASH. In the following research, we will focus on the study of secondary metabolites using activity tracking, amplification fermentation, and other approaches involving modification of the nutrient conditions in the medium and the genetic recombination of biosynthetic gene clusters.

## 4. Conclusions

A novel strain, NEAU-H2^T^, with antifungal activity was isolated from the rhizosphere soil of wheat (*Triticum aestivum L.*). Four compounds, including one new compound, along with three known congeners (3-Acetylamino-N-2-thienyl-propanamide, β-adenosine, 2-minaline), were isolated. Morphological and chemotaxonomic features together with phylogenetic analysis and genomes suggested that strain NEAU-H2^T^ belonged to the genus *Streptomyces*. Cultural and biochemical characteristics combined with DNA–DNA relatedness values clearly revealed that strain NEAU-H2^T^ was differentiated from its closely related strains. Based on the polyphasic taxonomic analysis, it is suggested that strain NEAU-H2^T^ represents a novel species of the genus *Streptomyces*, for which the name *Streptomyces triticiradicis* sp. nov. is proposed. The type strain is NEAU-H2^T^ (=CCTCC AA 2018031^T^ = DSM 109825^T^).

## Figures and Tables

**Figure 1 microorganisms-08-00077-f001:**
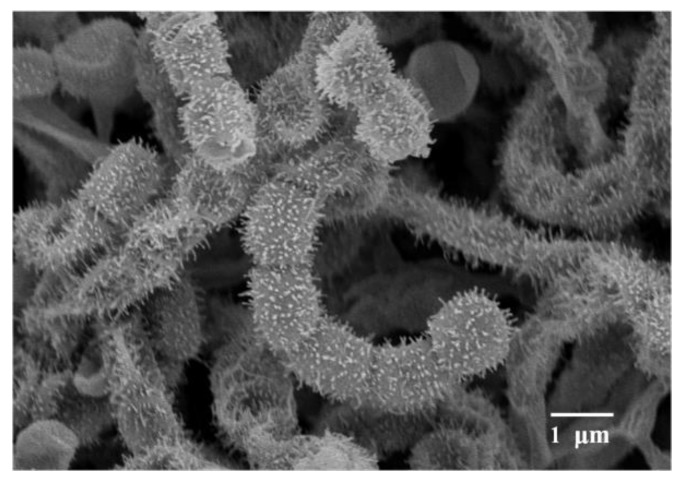
Scanning electron micrograph of strain NEAU-H2^T^ grown on International *Streptomyces* Project (ISP) medium 3 (ISP 3 ) for 2 weeks at 28 °C; Bar 1 μm.

**Figure 2 microorganisms-08-00077-f002:**
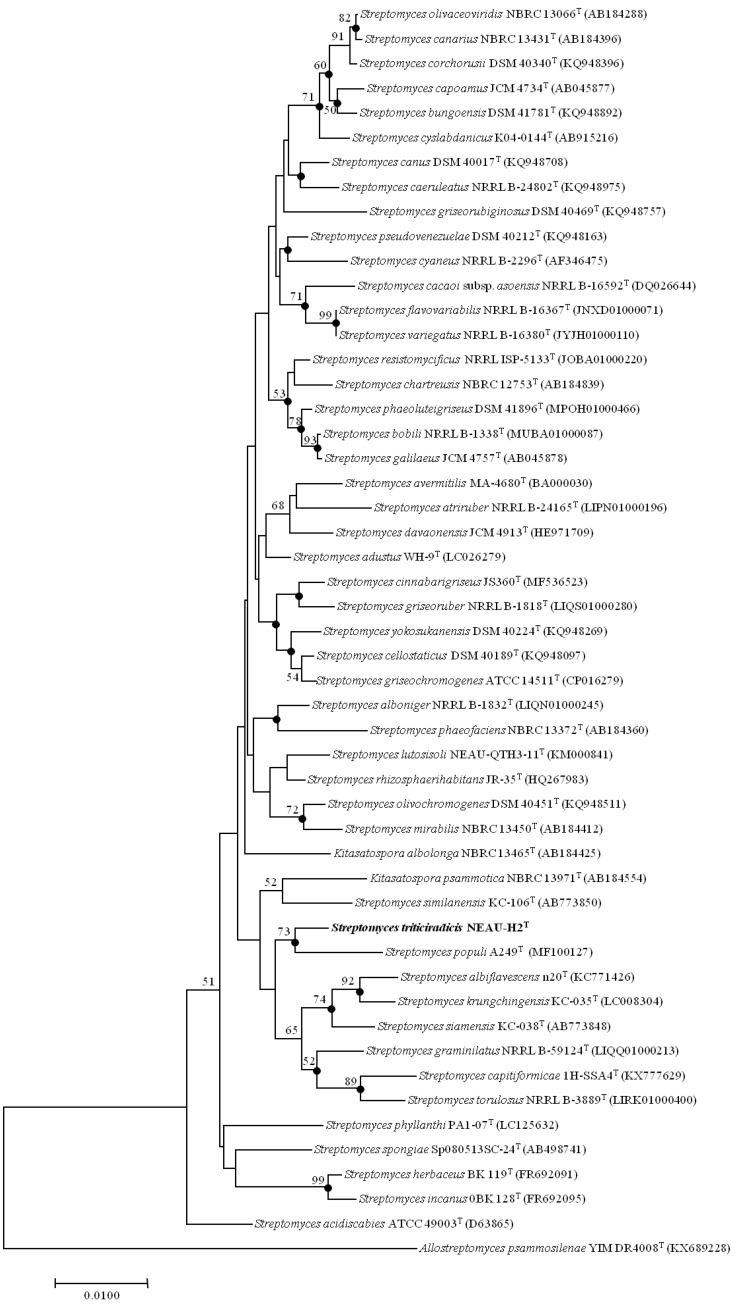
Neighbor-joining tree based on 16S rRNA gene sequences (1418 bp) showing the relationship of strain NEAU-H2^T^ (in bold) with related taxa, which are the top 50 type strains of *Streptomyces* species of gene sequence similarities based on analysis using EzTaxon-e. Filled circles indicate branches that were also recovered using the maximum likelihood methods. Only bootstrap values above 50% (percentages of 1000 replications) are indicated. *Allostreptomyces psammosilenae* YIM DR4008^T^ (KX689228) was used as an outgroup. Bar, 0.01 nucleotide substitutions per site.

**Figure 3 microorganisms-08-00077-f003:**
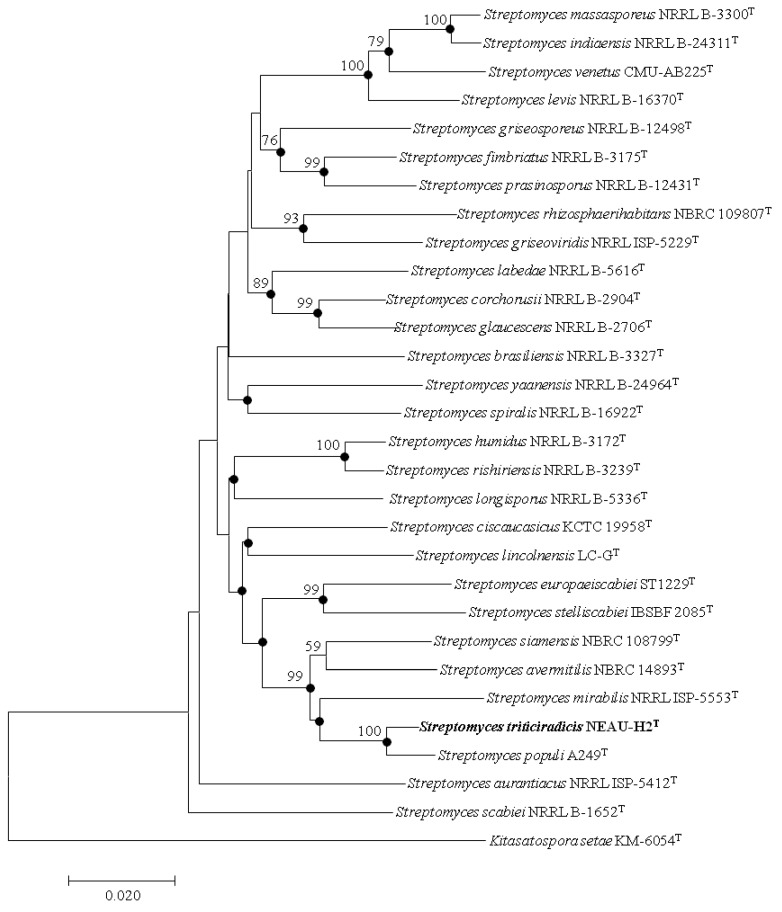
Neighbor-joining tree based on multilocus sequence analysis (MLSA)analysis of the concatenated partial sequences (2060 bp) from five housekeeping genes (*atpD*, *gyrB*, *recA*, *rpoB*, and *trpB*) of strain NEAU-H2^T^ (in bold) with related taxa. Filled circles indicate branches that were also recovered using the maximum likelihood methods. Only bootstrap values above 50% (percentages of 1000 replications) are indicated. *Kitasatospora setae* KM-6054^T^ was used as an outgroup. Bar, 0.02 nucleotide substitutions per site.

**Figure 4 microorganisms-08-00077-f004:**
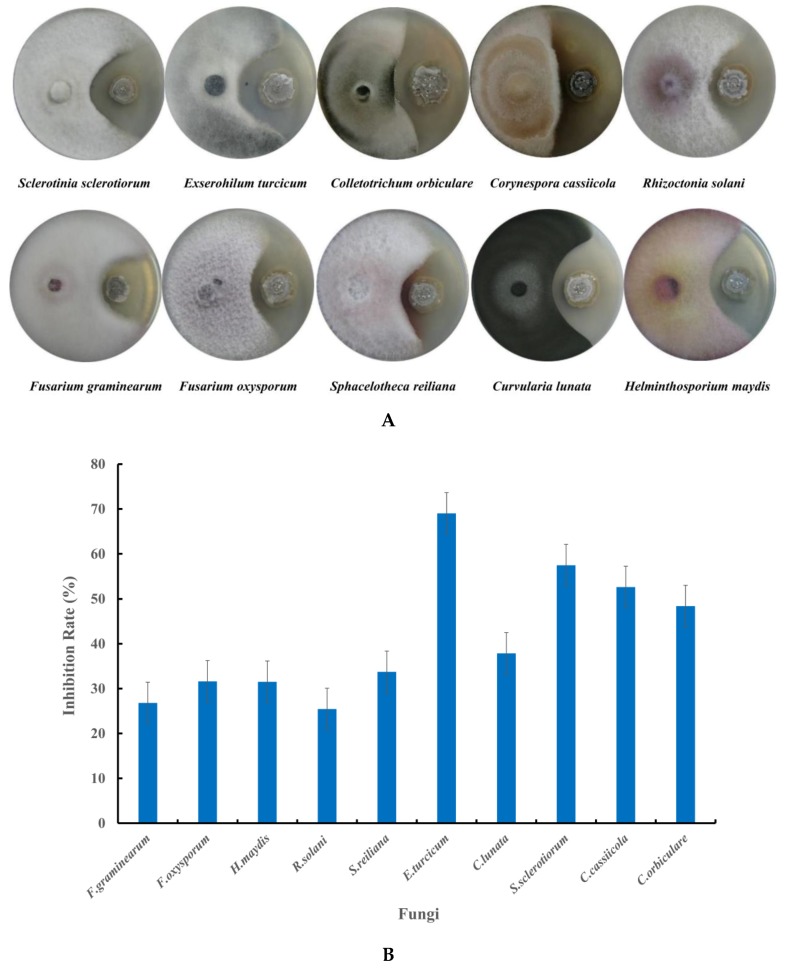
Antifungal activity of strain NEAU-H2^T^ against the tested fungi. (**A**) Dual culture plate assay against tested fungi; (**B**) Inhibition rate against the tested fungi.

**Figure 5 microorganisms-08-00077-f005:**
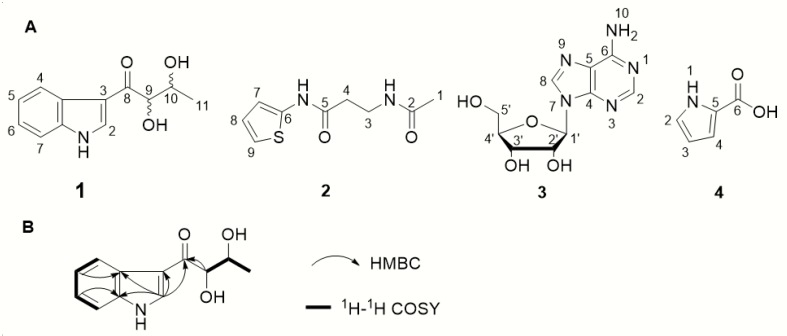
(**A**)The structure of compounds **1**–**4**; (**B**) 2D NMR correlations of **1**.

**Table 1 microorganisms-08-00077-t001:** Differential characteristics of strain NEAU-H2^T^ and its most closely related *Streptomyces* species. Strains: 1, NEAU-H2^T^; 2, *S. rhizosphaerihabitans* NBRC 109807^T^; 3, *S. populi* A249^T^; 4, *S. siamensis* NBRC 108799^T^. All data are from this study except where marked. +, positive; –, negative. ^‡^ Data from Lee et al. [48]; ^†^ Data from Wang et al. [49]; ^§^ Data from Sripreechasak et al. [50].

Characteristic	1	2	3	4
Spore chain	Spiral	Straight ^‡^	Straight ^†^	Spiral ^§^
Spore surface	Spiny	Hairy ^‡^	Rough ^†^	Smooth ^§^
Growth temperature range (°C)	4–40	10–40	10–37	10–40
Growth pH range	5–10	5–11	6–12	5–11
NaCl tolerance range (*w*/*v*, %)	0–9.0	0–10.0	0–4.0	0–10.0
Cellulose decomposition	+	+	–	–
Gelatin liquefaction	–	+	+	–
Catalase production	–	+	+	–
H_2_S production	+	–	+	+
Milk coagulation	+	+	+	–
Nitrate reduction	–	–	+	–
Starch hydrolysis	+	–	–	–
Tween 20 hydrolysis	+	+	+	–
Tween 80 hydrolysis	+	+	–	+
Nitrogen source utilization				
l-serine	–	+	+	+
l-threonine	+	+	–	+
l-tyrosine	–	+	+	+
Carbon source utilization				
l-arabinose	–	+	+	+
Dulcitol	–	–	–	+
*meso*-inositol	–	+	+	+
d-mannitol	–	+	–	+
l-rhamnose	+	–	+	–
d-ribose	–	+	–	+
d-sorbitol	–	+	–	+
d-xylose	–	+	+	–
Phospholipids *	DPG, PE, PL, PI, PIM	AL, DPG, GL, PE, PG, PI, PIM, 2PLs ^‡^	AL, APL, DPG, 2Ls, PE, PL, PIM ^†^	DPG, PE, PG, PI, PL ^§^
Menaquinones	MK-9(H_6_), MK-9(H_8_), MK-9(H_4_)	MK-9(H_6_), MK-9(H_8_) ^‡^	MK-9(H_6_), MK-9(H_8_), MK-9(H_2_), MK-9(H_4_) ^†^	MK-9(H_6_), MK-9(H_4_), MK-9(H_8_) ^§^
Whole cell-wall sugars	Glucose	Glucose, ribose ^‡^	Xylose, galactose ^†^	ND

* APL, aminophopholipid; DPG, diphosphatidylglycerol; PE, phosphatidylethanolamine; PI, phosphatidylinositol; PIM, phosphatidylinositol mannoside; AL, unidentified aminolipid; GL, unknown glycolipid; L, unidentified lipid; PL, unidentified phospholipid; MK, menaquinone; ND, no detection.

**Table 2 microorganisms-08-00077-t002:** ^1^H (600 MHz) and ^13^C (150 MHz) NMR Data of **1** in CD_3_OD.

No.	*δ* _C_	*δ*_H_ (mult, *J* in Hz)	^1^H-^1^H COSY	HMBC (H→C)
2	135.8	8.29 (s, 1H)		C-3, 3a, 7a, 8
3	116.3			
3a	127.3			
4	122.9	8.27 (d, *J* = 7.3 Hz, 1H)	H-5	C-6
5	123.4	7.21 (td, *J* = 7.2, 1.2 Hz, 1H)	H-5, H-6	C-4, 6, 3a, 7
6	124.5	7.24 (td, *J* = 7.2, 1.2 Hz, 1H)	H-5, H-7	C-4, 7a
7	112.9	7.46 (d, *J* = 7.4 Hz, 1H)	H-6	C-5, 3a
7a	138.2			
8	197.2			
9	79.7	4.74 (d, *J* = 4.8 Hz, 1H)	H-10	C-11, 10, 8
10	71.1	4.10 (m, 1H)	H-11, H-9	C-11, 8
11	17.9	1.16 (d, *J* = 6.4 Hz, 3H)	H-10	C-10, 9

* *δ*_C or H_: chemical shift; *J:* coupling constant; COSY: correlated spectroscopy; HMBC: ^1^H detected heteronuclear multiple bond correlation.

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
