# Peer review of "Taxonomic Characterization, and Secondary Metabolite Analysis of Streptomyces triticiradicis sp. nov.: A Novel Actinomycete with Antifungal Activity"

_microorganisms, 2020, doi:10.3390/microorganisms8010077_

Round 1

Reviewer 1 Report

The manuscript involves a taxonomic description of a new species  of Strptomyces and is proposed as Streptomyces triticiradicis and the identification of four compounds produced in liquid culture by the strain.

The data presented comply with all the chemotaxonomic analysis required to fullfil the taxonomic description. 

On another aspect, there is key information missing in the work presented.

The strain is presented a exhibiting  antifungal activity on agar medium against different pathogens. Surprisingly, when the authors attempt the characterization of this antifungal activity by culturing the strain in liquid medium, they choose only one single rich carbon medium, instead of testing alternative nutritional  conditions to reproduce the actvity observed in solid medium. On top of that no results are shown about the production of an antifungal activity in the fermentation or in the first crude extract, and they do not explain why they decide to isolate the compounds described in the manuscript, that finally  do not show any activity.

Furthermore, from the fragmented whole genome sequence data used, they suggedt the presence of a clsuter that could be associated to natamycin, but this hypothesis is not confirmed by testing by LC-MS the presence of known compoind in the different extract and partitions done.  

More specifically, the authors should improve and provide better references to cite the work in the introduction;

in the materials and methods section, no description is provided about the methods used to obtain high quality genomic DNA and the whole genome sequence that is used in the AntiSmash annotation analysis and that is described in lines 246-249.

the profiles in secondary metabolites of the strain extracts are not described

the sentence in line 227-228 should be rephrased. The "EZBiocloud analysis suggests  that the strains belong to the genus Streptomyces."

the description of the strain includes in the text the surface ornamentation (line 254-2559and a Figure 1 obtained from electron microscopy. There is no information included about the method used to obtain this description and it should be included in the materials and method section.

Figures in the text do not appear in order, Figure 5B is cited before Figure 4

there is not description of the test method used to evaluate the antifungal activity of the purified compounds

sentence in line 334 should be changed to: " in oder to further discover the biosynthetic potential of the strains, we performed..."

the section describing the genes identified with antismash is not relevant given the poor quality of the genome sequence with a large number of contigs , and there is no correlation established  with the known compounds produced in the liquid fermentation that are not characterozed by LC-MS against known libraries of mnatural products. This information should be included.

the authors cannot estbalish a conclusion from their study regarding the lack of active compounds obtained given that they do not test the activity of their crude fermentation prior to fractionation. in addition, the production conditions are not studied to ensure that they have manage to reproduce in liquid condition the production of the active molecules obsreved when the strain in grown on solid media. Sections from l-344-348 should be rephrase accordingly.

in paragraph from lines 349-352, additional analytical data need to be provided to confirm the presence or absence of the antifungal in the extract.

As no data are presented regarding the confirmation of the production of the antifungal activity  in the liquid medium, the authors cannot argue that more studies are necessary to trigger the production.

Reviewer 2 Report

The manuscript “Taxonomic Characterization and Secondary Metabolite Analysis of Streptomyces triticiradicis sp. nov., a Novel Actinomycete with Antifungal Activity” by Yu et al. describes the polyphasic approach adopted in the characterization of a novel actinomycete. Morphological, phenotypic, genotypic, chemotaxonomic and biochemical analyses were combined together to demonstrate the differences between the novel actinomycete strain NEAU-H2T and its closely related strains. A preliminary analysis, which includes structural characterization, of four isolated secondary metabolites from this strain has also been reported. An attempt has been made to look for biosynthetic gene clusters in the whole genome using antiSMASH.

The authors have presented several data supporting their claim. However, there are certain specific issues that can be further addressed:

Section 1: Introduction

Many of the sentences in introduction were framed in a grammatically incorrect way. For example, the sentence in lines 76-79 will need to be reviewed to convey the intended message. This is not an isolated example, and it would be in the best interest of the authors to review the introduction for obvious grammatical and typographical errors. The authors make a claim in lines 80-81 that “strain NEAU-H2Twith broad inhibitory activity against phytopathogenic fungi was isolated”. However, no detailed explanation has been given by the authors supporting their claim of the strain’s broad inhibitory activity in the results and discussion section. A more direct discussion regarding what is meant by broad inhibitory activity and how it is being displayed by the new strain should be included.

Section 2.1: Isolation of Actinobacterial Strain

In the section 2.1, a significantly more detailed explanation of the methodology adopted for the isolation of the bacterial strain should be included. This is would be beneficial for reviewers/readers. The authors have used the standard dilution plate method without the citation of any references for the same. This can further be seen in the following sections wherein some “standard” procedure has been used without any references to trace back and follow in setting up the experiment.

Section 2.2: Morphological and Biochemical Characteristics

A more detailed explanation of the color determination method would be better appreciated by the readers. For example, if it is the color of the substrate mycelium, the aerial mycelium or both. For growth experiments ISP 3 agar medium was chosen, without explaining the rationale behind using that media.

Section 2.3: Chemotaxonomic Analysis

The language of the entire methodology was same as a previous paper published (with some of the authors in both these papers). DOI for the paper: 10.1099/ijsem.0.002207.

Section 2.4: Phylogenetic Analysis

The methodology for total DNA extraction does not include the information about the amount of biomass started with. The details regarding the parameters used for generating the sequence similarity between strains is not reported. This is an important information, as in the results and discussion section, the authors compare the various polyphasic parameters between these “closely related” strains to show that their strain is novel and different from these closely related streptomyces strains.

Section 2.5: DNA-DNA Relatedness Tests

The genome mining experiment has been condensed in this section with just one sentence. In the reviewer’s opinion a more detailed explanation was needed here. Explaining in detail the isolation of DNA in the previous section would serve meaningfully for the genome mining analysis as well. The table S3 in the supplementary section has many information reading the genome sequence feature of two strains without any information regarding the approach adopted to get those values. These techniques can be included in a new section for genome mining methodology.

Section 2.6: In Vitro Antifungal Activity Test

No rationale has been provided for using the 10 phytopathogenic fungi and neither is any reference cited justifying the experimental motivation in selecting the fungal strains. Figure 5(B) shows the inhibition rate. However, no explanation is given in this section showing how these values were calculated.

Section 2.7: Isolation and Characterization of Secondary Metabolites

There is a claim by the authors that both the supernatant and the pellet extracts have the same HPLC and TIC analyses. However, no data supporting this claim is provided either in main text or supplementary file. HPLC traces should be included for this analysis. The methodology used for semipreparative scale isolation of metabolites should be explained in detailed (with the solvent gradient used for isolation also included).

Section 3.1: Polyphasic Taxonomic Characterization of NEAU-H2T

The authors have used several polyphasic characteristics as signature barcodes for identifying the genus Streptomyces, without citing references wherein these features have been shown to be unique to Streptomyces. Do the growth conditions, colony colors and other factors conform to the literature-based data? Without this intellectual input, there is no reason to have just the growth conditions listed. Here again, an extensive comparative analysis has been shown for the new strain and some other “closely related” strains, without prior information/rationale on why and how these particular strains were selected? For the cellular fatty acid profile and the predominant menaquinones, a comparison between this strain and others and a plausible insight into the similarity/differences would be informative.

Section 3.3: Antifungal Activity Evaluation

The authors claim that their data displays significant inhibitory effects against some fungal strains. However, no threshold value from literature has been discussed here to validate their claim.

Section 3.4: Identified of Secondary Metabolites from Strain NEAU-H2T

Grammatical error in the heading of the section. HPLC traces for the extracts from the strain and the negative controls should be included in the supplementary. The 2D NMR correlations can be explained in a more detailed way to validate their structure.

Section 3.5: Identification of Genes Responsible for Antifungal Property

The motivation for this section should be discussed in this section. Why is it required to use antiSMASH to discover biosynthetic gene clusters? It is definitely a good indication if the strain has many modular PKS and NRPS clusters, by why is it good/important has not been discussed. They bring the comparison to natamycin in this section. Have they tried to do a sequence similarity analysis between their strain and the strain producing natamycin? If yes, then what plausible reasons could be inferred?

In conclusion, there is a lot of data represented in this manuscript, but in many sections the intellectual input in analyzing the data is lacking. More detailed explanations are needed with proper references from literature, especially when comparison studies are done. Also, in a more broad and general perspective, the secondary metabolite analysis is not very meaningful. More detailed metabolomics analysis can be adopted, such as molecular networking to map the global metabolite production profile in the strain.

Round 2

Reviewer 1 Report

The authors have introduced the changes requested, but still no new data are provided regarding the new experiment needed to purify the antifungal compound. If the activity cannot be produced in liquid medium (the crude extract is not checked for its antifungal activity and we cannot assess if the strain produces any activity in liquid medium) , at least the authors should attempt to identify the components produced by the strain grown in solid agar medium and attempt their isolation.

Sections the following sectons still need an improvement and/or being edited:

Lines 73-74:  replace by: “because their novel scaffolds can provide new modes of action “

Lines 75-78:  from “Streptomyces….diverse biological activities” rephrase and improve sentence;

Starting with “Members of the genus Streptomyces….

Line 80: delete “such as mycostop”

Line 85: delete “In addition”, as there is not consequence of what is mentioned before.

Paragraph lines 391-397: can be deleted, there is nothing added of interest related with the discussion; it is normally expected to have many PKS and NRPS in Streptomyces: given the high number of contigs, many of these are only part of larger BCGs and they show only fragmented

Line 346: Include at the begining of the paragraph: “ Only major components were identified from the liquid fermentation extract.